# Update on Nutritional Advice Post-Heart Transplant: A Cross-Sectional Study across French-Speaking European Centers

**DOI:** 10.3390/nu16172843

**Published:** 2024-08-25

**Authors:** Jean-Baptiste Bonnet, Claire Trémolières, Clémence Furic-Bego, Laetitia Galibert, Ariane Sultan, Vincent Attalin, Antoine Avignon

**Affiliations:** 1Nutrition Diabetes, Transversal Nutrition Unit, University Hospital of Montpellier, 34295 Montpellier, France; 2UMR 1302, Institute Desbrest of Epidemiology and Public Health, University of Montpellier, INSERM, CHU, 34295 Montpellier, France; 3PhyMedExp, University of Montpellier, INSERM U1046, CNRS UMR 9214, 34295 Montpellier, France

**Keywords:** heart failure, transplantation, undernutrition, cardiovascular risk, foodborne infections

## Abstract

Introduction: Heart transplantation is the standard treatment for severe heart failure. Graft preservation and infectious risk secondary to immunosuppressive drugs lead healthcare teams to offer nutritional advice to patients upon discharge from the hospital. However, no consensus or recommendation is available. Method: We conducted a study to review the practices in all 26 centers providing heart transplantation in French-speaking Europe. We requested and analyzed the written documents these centers provided to their patients. The same two dieticians categorized the highlighted pieces of advice into distinct, autonomous categories. Results: We identified 116 pieces of advice, categorized into three areas: dietary restrictions for immunosuppressant/food interaction; environmental and food preparation guidelines and prevention of foodborne infections; and healthy and active lifestyle recommendations. Except for advice on immunosuppressant/food interaction, over one-third of the centers suggest discontinuing advice within 2 years post-transplant. General dietary advice covers lipids, carbohydrates, protein, calcium, sodium, and fiber but offers limited guidance on fatty acids despite their importance in cardiovascular risk prevention. Conclusion: This study represents a pioneering exploration of the nutritional advice provided to patients following cardiac transplantation. It underscores the critical necessity of establishing consensus-based clinical guidelines in this domain.

## 1. Introduction

In 2020, 370 patients in France underwent heart transplantation [1], the most effective treatment for severe heart failure [2]. While this life-saving procedure offers hope to patients with end-stage heart failure, it also introduces significant challenges in post-transplant care, particularly in the realm of nutrition. Following transplantation, several factors—including post-transplant weight gain [3], the onset of metabolic syndrome [4], and the use of immunosuppressive drugs and corticosteroids to prevent allograft rejection—prompt healthcare teams to provide targeted nutritional advice to patients upon discharge.

In addition to the need for weight management, there are several key concerns in post-transplant nutrition, including the prevention of foodborne infections due to immunosuppression, reduction of cardiovascular risk, and prevention of post-transplant diabetes [5]. Undernutrition is another crucial aspect that must be considered, especially following such a major surgical procedure [6]. Addressing these nutritional challenges is vital to improving patient outcomes and reducing the risk of complications.

Heart transplant recipients face unique vulnerabilities because of the immunosuppressive therapy required to prevent graft rejection. This therapy weakens the immune system, making patients highly susceptible to infections, including those from foodborne pathogens. Moreover, there is an elevated risk of adverse drug-food interactions, such as those with grapefruit, which can dangerously increase blood levels of immunosuppressants. These complexities highlight the critical need for precise and individualized nutritional guidance post-transplantation.

Despite the importance of nutrition in post-transplant care, to the best of our knowledge, there is no European consensus or recommendation regarding nutritional and dietary care following heart transplantation or transplantation of any other solid organ. Moreover, only limited data are available concerning nutrition and healthy lifestyle advice post-transplantation [7]. Notably, the guidelines from the International Society of Heart and Lung Transplantation for heart transplant recipients reveal that most of their recommendations are based on a C level of evidence, indicating reliance on expert consensus rather than randomized trials [5].

While many general hygiene and dietary practices are applicable across populations, heart transplant patients require more specific and stringent guidelines to address their increased risk of infection and to ensure the effectiveness of their medical treatments. However, there is a lack of harmonized, evidence-based dietary protocols across transplant centers, leading to considerable variability in care. This underscores the urgent need for standardized, evidence-based nutritional guidance tailored to the heart transplant population to optimize their recovery and long-term outcomes.

In 2011, the United States Department of Agriculture published a guide on post-transplant health safety to reduce the risk of foodborne infection [8]. In 2014, a study conducted in the United Kingdom and the Republic of Ireland examined nutritional practices following solid organ transplantation, revealing significant disparities between centers [9]. The study found wide variations in dietary instructions, particularly regarding the duration of dietary restrictions, and reported few instances of foodborne infections.

### Objective

Given the critical importance of short- and long-term nutritional and dietary interventions for the management of heart transplant patients, as evidenced in previous studies [7,10,11], our study aimed to comprehensively review the practices of heart transplant centers in French-speaking Europe.

## 2. Methods

### 2.1. Study Design

We conducted a cross-sectional study to analyze written nutritional recommendations given to patients following heart transplantation in French-speaking European centers.

### 2.2. Setting

A dietician from the nutritional unit of the University Hospital of Montpellier contacted, via phone, the nutritionists responsible for heart transplant patients at 26 centers, requesting them to send their discharge documents containing nutritional advice. Follow-up and rehabilitation care centers were excluded from this study. The research was conducted from October 2021 to February 2022.

### 2.3. Participants and Study Size

All 26 state-run centers in French-speaking Europe that perform heart transplants for adults were contacted and included in the study.

### 2.4. Variables

We collected data on the duration of these recommendations, as well as on key aspects of each center, including the presence of a dietician and the center’s volume of activity.

### 2.5. Date Sources

We requested all written documents that were provided to patients. However, documents produced internally at the center and not given to patients were excluded from our analysis. Centers had the option to send these documents by e-mail, fax, or post mail.

We orally inquired about the presence of a dietician dedicated to this activity at each center. Data regarding the center’s transplant activities were obtained from official activity reports.

### 2.6. Bias

Advice is often given orally, though this can be inconsistent. Therefore, we chose to concentrate exclusively on written advice provided to patients, considering that this form is most likely to be retained by patients after a major surgery such as heart transplantation.

We focused exclusively on transplant centers, excluding rehabilitation centers from our study. This decision was based on the fact that patients may be referred to various rehabilitation centers from the same transplant center, depending on individual circumstances.

All documents were meticulously analyzed by two dieticians. Any discrepancies encountered during the analysis were discussed collectively in a group, which included a physician. We approached this analysis without any preconceived expectations regarding the advice.

### 2.7. Analytical Methods

The advice was categorized into distinct, ideally self-contained sections, arranged in a logical sequence by a team comprising two dieticians and two physicians. We then investigated whether any centers omitted advice in any of the established categories. Additionally, we examined whether the provision of advice was influenced by the presence of a dietician or the center’s level of activity.

## 3. Results

### 3.1. Participants

All 26 centers responded to the evaluation (Appendix A).

### 3.2. Descriptive Data

All centers except one allocated time for a dietician. The number of heart transplants performed ranged from 4 to 90, with four centers completing fewer than 10 transplants and five centers completing more than 30. The median number of transplants was 14 per year.

### 3.3. Outcome Data

Through the analysis of all documents, we were able to identify a total of 116 different pieces of advice potentially given to patients (Appendix A). Among these, we observed two distinct sets of recommendations based on their prevalence across the centers.

The first set, endorsed by more than 75% of the centers, includes seven specific pieces of advice: (i) avoiding grapefruit; (ii) avoiding raw meat, fish, eggs, and cold cuts; (iii) avoiding raw, unsterilized, or pasteurized cheese, milk, dairy products; (iv) practicing good hand hygiene; (v) respecting the cold chain; (vi) adhering to expiration dates; and (vii) limiting simple carbohydrate intake. 

The second set, recommended by 50% to 75% of the centers, consists of nine separate pieces of advice including (i) avoiding smoked products, (ii) avoiding raw shellfish and mollusks, (iii) avoiding St. John’s wort, (iv) washing kitchen environment with bleach water; (v) avoiding cut-up food; (vi) maintaining environmental hygiene and safe food preparation practices; (vii) following a “balanced diet”; (viii) controlling lipids and (ix) controlling salt intake.

### 3.4. Main Results

Through thematic analysis, we categorized the 116 items into three main groups: (i) dietary restrictions to prevent interaction between immunosuppressants and food; (ii) advice on environmental hygiene, food preparation, and dietary measures to prevent foodborne infections; and (iii) recommendations for a healthy and active lifestyle. Additionally, we observed variability among the centers in terms of specifying the duration for which their recommendations should be applied.

### 3.5. Food Restrictions to Prevent Interaction with Immunosuppressants 

Twenty-four (92%) centers mentioned food exclusions related to the introduction of immunosuppressive therapies. Thirteen different foods were prohibited (Figure 1). Grapefruit was cited by the 24 centers, and 7 of the 13 prohibited foods are citrus fruits.

### 3.6. Advice Regarding Environment, Food Preparation, and Prevention of Foodborne Infections

A total of 24 centers (92%) reported providing advice regarding the environment, including food preparation, such as advice regarding the kitchen environment, food sourcing, storage, and cleanliness. The key advice for the prevention of foodborne infections was to avoid raw foods (Figure 2).

### 3.7. Advice on a Healthy and Active Lifestyle

Twenty centers (77%) reported providing advice on a healthy and active lifestyle, including general advice regarding a “balanced diet” (*n* = 18, 69%), regular physical activity (*n* = 7, 27%), avoiding any alcohol consumption (*n* = 7, 27%), and enriched food to fight against undernutrition (*n* = 3, 12%). General dietary advice included advice on lipids, carbohydrates, protein, calcium, sodium, and fiber. However, specific recommendations regarding daily caloric intake were notably absent or inconsistent across the majority of centers, with only three centers (12%) providing such guidance.

The advice associated with lipids included controlled intake of total lipids (*n* = 14, 54%) and limitation of saturated fats (*n* = 7, 27%), but only one center addressed the issue of omega-3 fatty acid enrichment.

The topic of carbohydrates focused primarily on a reduction in sweet products (*n* = 20, 77%), limiting fruits (*n* = 12, 46%), and regulating complex carbohydrates (*n* = 8, 31%). For protein intake, the centers suggested one to two portions of protein per day (*n* = 8, 31%). The advice on calcium (*n* = 9, 35%) was in line with the American nutritional recommendations for transplant patients, regardless of the type of transplant (i.e., 1000 to 1500 mg daily intake) [8].

Nineteen centers (73%) advised to control salt intake with widely varying advice ranging from 2 to 8 g per day. Eight centers (31%) recommended salt restriction without quantitative benchmarks (Appendix A).

### 3.8. Period of Application of the Recommended Measures

Among the centers that advised food avoidance to prevent foodborne illness (*n* = 24), duration of application was only provided in 14 centers (58.3%) (Figure 3). This duration ranged from 2 months to 1 year. Food avoidance to prevent interaction with immunosuppressive drugs did not include duration for any of the centers. 

### 3.9. Analysis by Center Specificity

Only one center did not have a dietician, and, notably, this was the same center that provided no written advice. Another center provided advice exclusively for the period surrounding the intensive care unit stay, as their dietician was based in this department. These two centers, which perform the fewest transplants annually, illustrate how resource availability can impact advice dissemination. Beyond these instances, the distribution of category-specific advice did not correlate with the number of transplants performed by each center.

## 4. Discussion

### 4.1. Key Results

This study represents the first comprehensive cross-sectional analysis of nutritional advice given to heart transplant patients in French-speaking Europe. We identified a diverse array of 116 different pieces of advice, categorized under three main themes: (i) dietary restrictions to prevent interaction with immunosuppressants, (ii) guidelines on environmental hygiene, food preparation, and prevention of foodborne infections, and (iii) recommendations for a healthy and active lifestyle. Notably, the study revealed a significant variability in the frequency and type of advice given across the 26 centers. While some recommendations were consistently shared by most centers, others varied considerably, highlighting the lack of uniformity in nutritional guidance post-heart transplantation. Additionally, we observed disparities in the advised duration for following these recommendations, reflecting different clinical practices and approaches across the centers. This variability underscores the need for standardizing nutritional advice to ensure consistent and optimal patient care in the post-transplantation period. Moreover, it is important to recognize that direct evidence linking these recommendations to specific clinical outcomes, such as reduced graft rejection or lower infection rates, is limited. Current advice is largely based on expert consensus aimed at minimizing risks associated with immunosuppressive therapy.

### 4.2. Strengths

This study benefits from a comprehensive response rate, as all heart transplant centers in French-speaking Europe were contacted, and each provided a response, ensuring a high level of representativeness and reliability in our findings. The focus on a specific linguistic and cultural region minimizes translation bias and ensures consistency in the interpretation of nutritional advice, as these professionals often share a common scientific background within “French-speaking” scientific societies. Furthermore, the analysis of the data was conducted by two qualified dieticians, adding a layer of specialized expertise to our evaluation. Notably, the study was designed independently of other existing nutritional recommendations, allowing for an unbiased and original exploration of current practices. The diversity of the 116 different pieces of advice identified, categorized under three main themes, and the detailed analysis of these practices across a broad geographical area underscore the thoroughness and depth of our research, making it a significant contribution to the field of post-transplant care. 

### 4.3. Limitations

Our study focused solely on written documents, which poses a significant limitation as a considerable amount of advice may also be imparted orally in a systematic manner. Therefore, some potentially valuable guidance might not have been captured in our analysis. Additionally, we did not include documents from rehabilitation centers, which might offer further insights into the nutritional advice provided during the crucial recovery phase. Our reliance on the self-declaration of the centers for the provision of documents introduces a potential bias, as we cannot verify the completeness of the documents received or their consistent distribution to patients. Moreover, the geographical scope of our study was limited to French-speaking European centers, potentially limiting the generalizability of our findings to other linguistic or cultural contexts. Lastly, the initial data collection was conducted in French, and the subsequent translation into English for publication purposes could introduce a potential bias in the interpretation of results. This linguistic transition may affect the nuanced understanding of the advice and recommendations. Furthermore, our study did not assess the direct impact of nutritional advice on patient outcomes, which is a crucial aspect for understanding the effectiveness of these recommendations in the post-transplant care process. Furthermore, we did not assess patient compliance with the dietary advice, which is crucial for understanding the real-world effectiveness of these recommendations in improving clinical outcomes. Future studies should aim to evaluate both compliance and its impact on patient health and graft survival.

### 4.4. Interpretation

Our findings indicate a partial adoption of the International Society of Heart and Lung Transplantation guidelines by the French-speaking centers, highlighting a notable variability in the application of these guidelines. Interestingly, the scope of topics covered in the written documents from these centers is broader than the guidelines, with a significant emphasis on the prevention of foodborne infections, which is not as prominently featured in international guidelines.

The consistent advice across centers to avoid grapefruit and the cautious approach towards St. John’s wort due to their interactions with immunosuppressive drugs underlines a widespread clinical consensus. Grapefruit, despite not being a staple in the daily diet of most European populations, can inhibit the cytochrome P450 3A4 enzyme (CYP3A4) and the P-glycoprotein transporter, significantly increasing blood levels of immunosuppressants and thus the risk of adverse effects [12]. This focus is aligned with the 2010 Guidelines for the Care of Heart Transplant Recipients [5], though it is worth noting that such specific restrictions are absent in the guidelines from the United States Department of Agriculture [8] and British counterparts [9], indicating regional variations in post-transplant dietary recommendations. Nevertheless, given the potential severity of drug interactions, even occasional consumption of grapefruit justifies its inclusion in these recommendations for transplant patients.

Immunosuppressive drugs increase the risk of infection after transplantation by 15–20% [9], and infectious diseases are one of the main causes of mortality after organ transplantation [13]. Particular attention is paid to certain pathogens, such as *Aspergillus*, *Listeria monocytogenes*, *Salmonella*, and *Streptococcus pneumoniae*. However, to our knowledge, no study has evaluated the rate of foodborne infections in post-transplant populations depending on whether the advice is followed or not. McGeeney et al. questioned the different English centers and found only rare cases of foodborne infections [9]. There has also been no evaluation of the impact of an eviction or any other hygienic advice on the population of post-transplant patients. This raises the question of the impact of the advice, which is not based on scientific evidence, on patients’ quality of life. If the precautionary principle is the reason for most of the advice given to patients, it is reasonable to question its legitimacy. Much of the advice is very similar to that usually given during pregnancy and is well-known by both medical professionals and the public [14].

We noted a wide variety of hygiene advice. Most are basic hygiene rules, especially for nutrition in a collective environment. Some of this advice represents a certain level of constraint in day-to-day life and can represent run costs. For example, the absence of cut products or bulk purchases can conflict with the implementation of a healthy lifestyle and should be evaluated. Nevertheless, the advice is less extensive than that suggested by the USDA [8]. Notably, more than half of the centers propose an end date to their advice within the year post-transplant (i.e., after beginning the dose reduction of immunosuppressive drugs).

We noted a wide disparity in advice for daily salt doses. This is consistent with a lack of specific recommendations during corticosteroid therapy. Nevertheless, three-quarters of the centers propose a reduction in the daily dose. However, a low-salt diet can be a source of undernutrition [15].

Previous findings specific to transplant patients [10], or more specifically to cardiac patients [5,7] on fatty acid, protein, or glucose intakes, have been very close to the recommendations for the general population [16], with evidence of effectiveness in metabolic diseases. In the context of long-term corticosteroid therapy and immunosuppressive treatment, the focus is on the prevention of new-onset diabetes after transplantation [17]. However, we were concerned about how little advice was given regarding fatty acid intake in view of the high level of evidence for the prevention of cardiovascular risk [18]. Moreover, while some centers focused on preventing malnutrition, specific caloric targets were rarely provided, which represents a gap considering the high risk of malnutrition in heart transplant patients. Future guidelines should incorporate specific calorie recommendations tailored to the patient’s metabolic status and recovery phase.

The significant variability in dietary advice observed across centers may indeed reflect a reliance on clinical experience, expert consensus, and common sense rather than on robust experimental evidence. While many recommendations, such as the avoidance of foods that interact with immunosuppressive drugs or pose a risk of foodborne infections, are grounded in sound clinical reasoning, the lack of large-scale, randomized trials evaluating these interventions creates an environment where common sense often guides practice. This variability underscores the need for more rigorous research to validate specific dietary recommendations and ensure they are based on scientific evidence rather than precautionary principles alone. Standardizing these guidelines will require a stronger foundation of experimental data, which is currently limited in the context of heart transplantation nutrition.

Prospective studies evaluating each piece of advice should be conducted. An attempt should also be made to measure the true rate of foodborne infections in this population in order to adjust and harmonize our advice. Such studies may be able to take into account the quality of life of the patients, as well as the preservation of the organ. Future studies should evaluate these nutritional recommendations prospectively, both in terms of compliance and clinical outcomes, such as infection rates and organ preservation. Assessing the true rate of foodborne infections in this population would be particularly helpful in refining and harmonizing these guidelines.

Our study revealed a broader range of advice compared to the previous UK study by McGeeney et al. [9]. However, most of the advice found in that study was also identified in ours. Interestingly, recommendations regarding probiotics and specific food evictions such as pre-packed salads, soft ice cream, fish eggs, various egg preparations, or blue-veined cheeses were not as prominent in our findings. These differences may reflect distinct cultural dietary habits. Notably, the near-systematic avoidance of pate in British centers, likely influenced by the history of the Creutzfeldt–Jakob disease, was not a focus of our study. Furthermore, we observed slight variations in naming, such as ‘salads from salad bars or delis’ compared to our finding of ‘raw vegetables outside the home’, highlighting potential challenges in translation across different linguistic contexts. Comparatively, the US documentation, although featuring a shorter list of food evictions, provides more detailed advice on worktop hygiene and cooking methods and does not specify an end date for these recommendations, leaving this decision possibly to the discretion of the transplant center [9].

### 4.5. Size of the Center

The size of the centers appeared to influence the allocation of dedicated dietetic time for this activity. Larger centers were not only able to extend their focus beyond the intensive care period but also more likely to produce written recommendations. This suggests that center size may play a role in the comprehensiveness of post-transplant care.

## 5. Conclusions

Our comprehensive cross-sectional study has illuminated the diverse landscape of nutritional advice provided to heart transplant patients across French-speaking European centers. We have unearthed a striking variability in the recommendations, underscoring a crucial need for more unified guidelines that can bridge the gaps between different centers and cultures. The discovery that some key dietary advice, such as the avoidance of certain foods due to interactions with immunosuppressants, is universally acknowledged, while other advice varies significantly, points to the complex nature of nutritional care in post-transplant scenarios. Although this study did not explore clinical outcomes in relation to dietary habits, it highlights the need for standardized and evidence-based dietary guidance to support consistent care practices across centers.

As we move forward, it becomes clear that there is an urgent need for prospective studies to evaluate the impact of these nutritional guidelines on patient outcomes. Such research would offer invaluable insights into optimizing post-transplant care and ensuring the highest quality of life for recipients. Our findings pave the way for future explorations and collaborations, aiming to establish a global consensus on post-heart transplant nutritional care that is both scientifically sound and culturally sensitive. In doing so, we hope to contribute to a future where post-transplant nutrition is not only a matter of clinical recommendation but also a cornerstone of successful patient recovery and long-term health.

## Figures and Tables

**Figure 1 nutrients-16-02843-f001:**
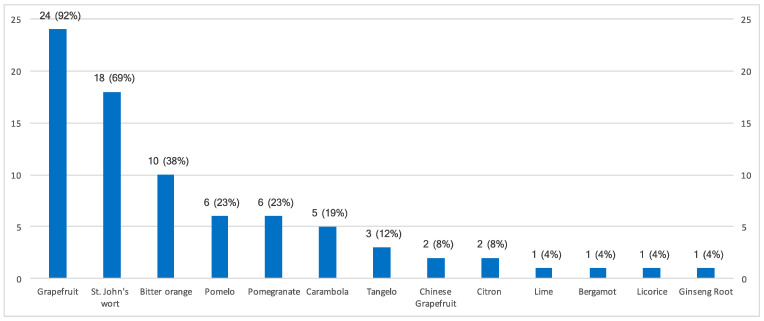
Number of centers with food avoidance related to immunosuppressive drugs (*n* = 26). Values are *n* (%).

**Figure 2 nutrients-16-02843-f002:**
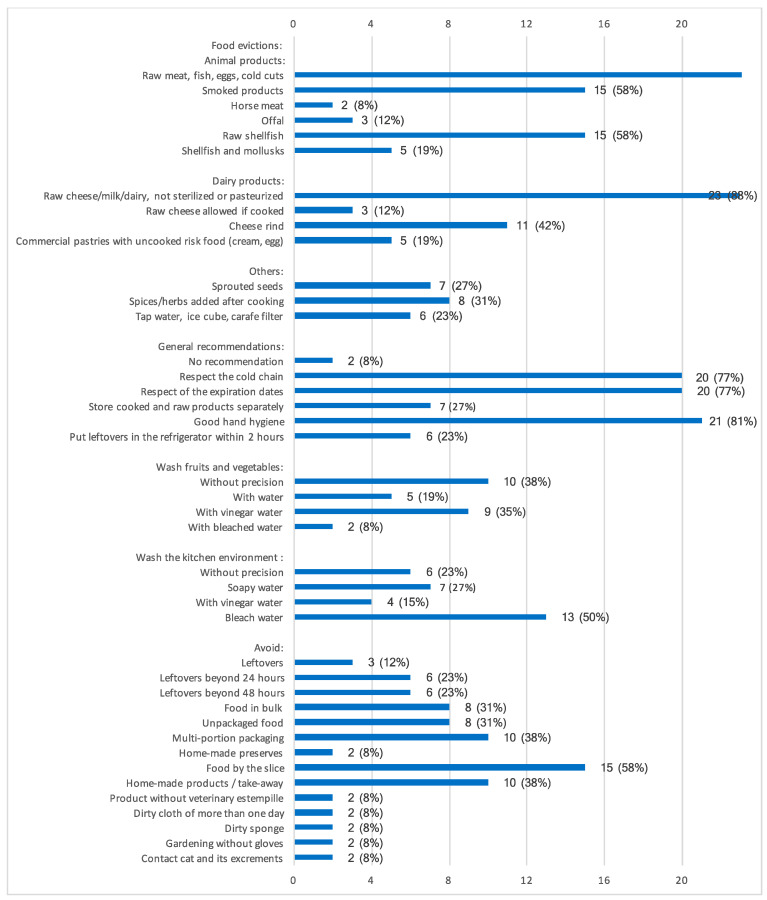
Advice on the environment, food preparation, and food evictions to prevent foodborne infections. Data are presented as number of centers that gave each advice. Values are *n* (%).

**Figure 3 nutrients-16-02843-f003:**
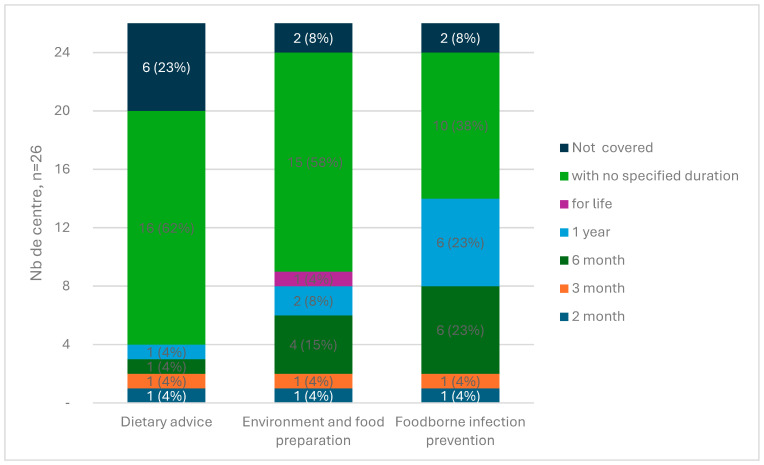
Duration of nutritional and health recommendations (*n* = 26).

## Data Availability

The datasets generated during and/or analyzed during the current study are available from the corresponding author upon reasonable request due to legal reasons.

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
