# Peer review of "Update on Nutritional Advice Post-Heart Transplant: A Cross-Sectional Study across French-Speaking European Centers"

_nutrients, 2024, doi:10.3390/nu16172843_

Round 1

Reviewer 1 Report

Comments and Suggestions for Authors

I find the topic of this contribution interesting, rigorous as regards the retrieval of the information and its completeness albeit somewhat restricted and rightly critical as regards the scientific soundness of several dietary proposals.

Some points to be clarified for the sake of the interested readers:

1.Avoidance of grapefruit has a degree of general consensus but I doubt that grapefruits are part of the daily diet of the majority of the european populations generally speaking and the French speaking transplanted population in particular.  Is it really useful?

2. Useful, in my opinion, to provide the reader with the validation of the dietary suggestion in terms of clinical outcomes, if existing, as well as the patient compliance to the dietary advices.

 3. Does diversity of advices among centres imply reliance upon common sense rather than experimental evidence?

Author Response

Dear Editor,

Thank you for the opportunity to revise our manuscript titled “Update on Nutritional Advice Post-Heart Transplant: A Cross-Sectional Study Across French-speaking European Centers.” We sincerely appreciate the insightful feedback provided by the reviewers. Their comments and suggestions have notably enhanced the quality and clarity of our report, and we have carefully addressed each point raised.

In response to the reviewers' feedback, we have made significant revisions to the manuscript, which we believe have strengthened both the narrative and the scientific rigor of our study. We have incorporated additional data, expanded certain discussions, and made clarifications where needed.

We are grateful to the reviewers for their constructive evaluation and thoughtful feedback. We believe that the revised manuscript now comprehensively addresses the concerns raised and offers a detailed analysis of nutritional advice for post-heart transplant care across French-speaking European centers.

We kindly request that you reconsider our manuscript for publication in light of these revisions. We are confident that our study provides valuable insights into patient care following heart transplantation and will be of great interest to the readership of Nutrients.

Thank you for your time and consideration. We look forward to your feedback.

Responses to reviewer 1:

Reviewer Comment:
Avoidance of grapefruit has a degree of general consensus, but I doubt that grapefruits are part of the daily diet of the majority of the European populations, generally speaking, and the French-speaking transplanted population in particular. Is it really useful?

Response: The inclusion of grapefruit in the list of foods to avoid for post-transplant patients is indeed based on a well-established clinical rationale. Grapefruit, along with other citrus fruits like pomelo and bitter orange, is known to interact with immunosuppressive drugs by inhibiting the cytochrome P450 3A4 enzyme (CYP3A4) and the P-glycoprotein transporter, which leads to increased levels of these drugs in the bloodstream and, subsequently, a higher risk of adverse effects. We added a reference to support this point (Nephrology (Carlton). 2008 Jun;13(4):337-47.)

While it is true that grapefruit may not be a staple in the daily diet of many Europeans, including French-speaking transplanted populations, its consumption remains common enough to pose a significant risk. Additionally, given that transplant recipients are a high-risk group, it is prudent to err on the side of caution with recommendations that prevent potentially life-threatening drug interactions.

Proposed Modification: We suggest keeping the advice to avoid grapefruit, but acknowledging in the discussion section of the article that while grapefruit may not be a significant component of many European diets, its well-documented interaction with immunosuppressive drugs warrants its inclusion in dietary recommendations for transplant patients. These modifications have been included in the 3rd paragraph, page 6.

Reviewer Comment:
Useful, in my opinion, to provide the reader with the validation of the dietary suggestion in terms of clinical outcomes, if existing, as well as the patient compliance to the dietary advice.

Response: Thank you for this insightful comment. We acknowledge that providing evidence of the clinical impact of dietary recommendations and information regarding patient compliance would enhance the relevance and utility of our findings. Unfortunately, direct evidence linking specific dietary recommendations to clinical outcomes, such as reduced mortality or lower rates of graft rejection, is scarce in the current literature, particularly in the context of heart transplant patients. Most of the recommendations, such as those related to foodborne infection prevention and immunosuppressant interactions, are based on expert consensus rather than large-scale clinical trials.

Proposed Modification: We added a section in the discussion acknowledging this gap in the literature and emphasize the need for future studies to assess the clinical impact of these dietary guidelines and patient adherence to them. Additionally, we integrated findings from available studies on compliance and its association with health outcomes, when applicable.

Text added in the Discussion section:

3rd paragraph page 5 :

Moreover, it is important to recognize that direct evidence linking these recommendations to specific clinical outcomes, such as reduced graft rejection or lower infection rates, is limited. Current advice is largely based on expert consensus aimed at minimizing risks associated with immunosuppressive therapy.

1stparagraph page 6 :

Furthermore, we did not assess patient compliance to the dietary advice, which is crucial for understanding the real-world effectiveness of these recommendations in improving clinical outcomes. Future studies should aim to evaluate both compliance and its impact on patient health and graft survival.

5th paragraph page 7 :

Future studies should evaluate these nutritional recommendations prospectively, both in terms of compliance and clinical outcomes, such as infection rates and organ preservation. Assessing the true rate of foodborne infections in this population would be particularly helpful in refining and harmonizing these guidelines.

Reviewer Comment:
Does diversity of advice among centers imply reliance upon common sense rather than experimental evidence?

Response:
Thank you for this important observation. Indeed, the diversity of advice we observed across different centers does suggest that, in the absence of strong experimental evidence, much of the dietary guidance given to transplant patients may be based on expert consensus, clinical experience, and common sense. This reflects a broader challenge in the field of transplantation nutrition, where rigorous, randomized controlled trials evaluating the impact of specific dietary interventions are scarce. Consequently, many recommendations are influenced by the precautionary principle, aiming to reduce risks associated with immunosuppressive therapy and post-transplant complications, even if these recommendations have not been extensively validated through experimental evidence.

Proposed Modification:
Page 6:

We added the 4th paragraph in this discussion section to acknowledge that the variability in recommendations may stem from the reliance on common sense and expert opinion in the absence of definitive experimental data. We also emphasize the need for future studies to provide an evidence base that can guide more standardized recommendations.

Reviewer 2 Report

Comments and Suggestions for Authors

This study aimed to assess nutritional advice provided post-heart transplant through a cross-sectional analysis of French-speaking European centers. While it highlights the lack of specific dietary recommendations for heart transplant patients, it does not provide sufficient evidence to draw definitive conclusions regarding the impact of dietary guidance on patient outcomes. Therefore, while the need for consensus-based guidelines is emphasized, the study's findings do not fully justify certain conclusions about the importance of dietary advice in this patient population.

Introduction, the authors should provide  a stronger justification for the need to formulate specific recommendations for heart transplant recipients. Most recommendations focus on basic hygiene practices that should be universally followed.

Results, given the risk of malnutrition in this patient population, were any specific recommendations regarding calorie intake provided?

Discussion, the concerns mentioned in the generalizability paragraph have already been addressed in the limitations section. Please remove them to avoid redundancy.

Conclusion, “This study not only highlights the importance of tailored dietary guidance”, as the study did not explore clinical outcomes in relation to patients' dietary habits, the importance of dietary guidance is not substantiated by the current findings. Please rephrase.

Comments on the Quality of English Language

Minor editing of English language required.

Author Response

Dear Editor,

Thank you for the opportunity to revise our manuscript titled “Update on Nutritional Advice Post-Heart Transplant: A Cross-Sectional Study Across French-speaking European Centers.” We sincerely appreciate the insightful feedback provided by the reviewers. Their comments and suggestions have notably enhanced the quality and clarity of our report, and we have carefully addressed each point raised.

In response to the reviewers' feedback, we have made significant revisions to the manuscript, which we believe have strengthened both the narrative and the scientific rigor of our study. We have incorporated additional data, expanded certain discussions, and made clarifications where needed.

We are grateful to the reviewers for their constructive evaluation and thoughtful feedback. We believe that the revised manuscript now comprehensively addresses the concerns raised and offers a detailed analysis of nutritional advice for post-heart transplant care across French-speaking European centers.

We kindly request that you reconsider our manuscript for publication in light of these revisions. We are confident that our study provides valuable insights into patient care following heart transplantation and will be of great interest to the readership of Nutrients.

Thank you for your time and consideration. We look forward to your feedback.

Responses to reviewer 2:

Reviewer Comment:
Introduction: The authors should provide a stronger justification for the need to formulate specific recommendations for heart transplant recipients. Most recommendations focus on basic hygiene practices that should be universally followed.

Response:
Thank you for this comment. We agree that the introduction could benefit from a stronger rationale explaining the need for specific nutritional recommendations tailored to heart transplant recipients. Although many hygiene practices are indeed universal, heart transplant patients face unique challenges due to their immunosuppressed status, which makes them particularly vulnerable to infections and complications that may not be as relevant to the general population. The immunosuppressive therapies required to prevent organ rejection increase susceptibility to foodborne infections and drug-food interactions, necessitating more detailed and specific advice for this patient group.

Proposed Modification:
We strengthened the introduction by expanding on the specific vulnerabilities of heart transplant recipients, particularly their heightened risk of foodborne illnesses and the crucial importance of preventing interactions with immunosuppressant drugs. Additionally, we highlighted the lack of standardized guidelines across centers, further underscoring the need for tailored recommendations. As a result, substantial changes were made throughout the introduction.

Reviewer Comment:
Results: Given the risk of malnutrition in this patient population, were any specific recommendations regarding calorie intake provided?

Response:
Thank you for this valuable comment. In our study, we observed that recommendations related to calorie intake were not consistently provided across the different centers. While some centers emphasized the importance of adequate nutrition to prevent malnutrition, particularly in the early post-transplant phase, explicit recommendations for calorie intake were limited to 3 centers. The focus was generally more on macronutrient balance, including protein, lipid, and carbohydrate intake, rather than on specific caloric targets.

Proposed Modification:
We added a sentence in the Results section (page 4) and additional elements in the Discussion section (page 7, 3rd paragraph) to address the lack of consistent recommendations for calorie intake across centers and to highlight the importance of addressing caloric needs, particularly given the high risk of malnutrition in heart transplant patients. We also emphasized that future guidelines should include specific calorie recommendations tailored to the patient's metabolic status and stage of recovery.

Reviewer Comment:
Discussion: The concerns mentioned in the generalizability paragraph have already been addressed in the limitations section. Please remove them to avoid redundancy.

Response:
Thank you for your valuable observation. Following your suggestion, we have removed the "Generalisability" paragraph from the Discussion section to avoid redundancy, as these concerns were already addressed in the Limitations section. We believe this revision improves the clarity and flow of the manuscript.

Reviewer Comment:
Conclusion: “This study not only highlights the importance of tailored dietary guidance,” as the study did not explore clinical outcomes in relation to patients' dietary habits, the importance of dietary guidance is not substantiated by the current findings. Please rephrase.

Response:
Thank you for your insightful comment. We agree that the original wording may have implied that our study demonstrated the impact of dietary guidance on clinical outcomes, which was not the focus of our research. We have revised the conclusion to clarify that, while we did not explore clinical outcomes in relation to dietary habits, the study highlights the need for standardized and evidence-based dietary guidance to ensure consistent care practices across centers. We believe this revision better aligns with the scope of our findings

Round 2

Reviewer 2 Report

Comments and Suggestions for Authors

I would like to thank the authors for considering my comments and appropriately revising their manuscript.